# Production of Soda Lime Glass Having Antibacterial Property for Industrial Applications

**DOI:** 10.3390/ma13214827

**Published:** 2020-10-28

**Authors:** Barış Demirel, Melek Erol Taygun

**Affiliations:** 1Sisecam Science, Technology and Design Center, 41400 Gebze, Kocaeli, Turkey; 2Department of Chemical Engineering, Istanbul Technical University, 34469 Maslak, Istanbul, Turkey; erolm@itu.edu.tr

**Keywords:** antibacterial, glass, soda lime, classical melting method, metal ions

## Abstract

This study was aimed to produce and characterize the first commercial glass materials with enhanced antibacterial property using conventional melting method. For this purpose, typical container glass composition that contains some specific metal ions, such as silver, strontium, and copper, was used to obtain antibacterial glass samples using classical melting method. After the melting process, antibacterial tests and migration tests were applied to the glasses, and it was found that the glass doped with 2% Ag_2_O was the best composition. X-rays diffractometer (XRD), thermal expansion coefficient, density, refractive index, hardness, and elastic module results showed that the glass doped with 2% Ag_2_O was a suitable material as a container glass. High Temperature Melting Observation System studies were performed on the produced antibacterial glass composition, and it was found that the antibacterial glass can be produced in soda lime glass furnaces without changing any furnace design and production parameters. As a result of the characterization studies, it was concluded that the produced container glass doped with silver can be a good candidate for food and pharmaceutical products where bacterial growth is absolutely undesirable.

## 1. Introduction

Glass is one of the most important material in human life. It can be used in different areas, such as windows, cars, home goods, food and drug packaging, with valuable features like being recyclable, easy to clean, durable, and easy to produce in different shapes and colors. Therefore, glass is an indispensable material in people’s daily lives [1].

On the other hand, the world population is increasing and at the same time, environmental pollution is becoming a bigger threat for humankind day by day. People contact with microorganisms in the ambient environment and this is one of the significant disease-causing factors. The factors such as changing living conditions, spending most of the time outside, changing eating habits, transport facilities, and international visiting cause to transfer microorganisms easily among individuals in public places and thus an increase in infectious diseases [2]. When the amount of microorganisms increases at a certain percentage, epidemics can occur depending on the intensity of infectious diseases. The well-being of individuals must be protected for a healthy society. Therefore, ensuring hygiene for the products used in the environment where we live and work, in other words, decontamination of microorganisms that can cause diseases has become important [3].

Scientists have been developing new methodologies to overcome this threat. There are many ways to struggle with bacteria and viruses. However, ensuring the presence of the ions, which have antibacterial effect, to prevent bacterial growth is an important one [4,5,6,7]. Some metal ions (i.e., silver, strontium, etc.) have functions to struggle bacteria and deactivate their enzymes, which are encouraged to be used in glass products [8,9,10,11,12,13]. There are many studies about ion-doped antibacterial bioactive glasses in the literature; however; none of these glasses are commercial glass products [14,15,16,17,18].

Some antibacterial commercial glass products have been found in the literature; however antibacterial property was gained to these materials using sol–gel coatings [19,20,21,22,23,24,25,26,27,28,29]. These coatings peeled out from the surface of the glass materials after a while and the glass materials lost their antibacterial property. However, antibacterial glass materials obtained by classical melting method eliminate these problems. Overall, this study aims to investigate, produce, and characterize the first commercial glass materials with enhanced antibacterial property produced by classical melting method.

## 2. Experimental Procedure

### 2.1. Glass Composition and Melting

In this study, initially, it was decided to add silver, strontium, and copper ions, which have very strong antibacterial effect, to soda lime glass. As a model soda lime glass, a typical glass container composition was selected as the glass composition due to the increasing importance of antibacterial properties for food industry (Table 1) [30].

Silver (I) oxide (Alfa Aesar, Ward Hill, MA, USA 99+%, metals basis), copper (II) oxide (Alfa Aesar, 99.0% min, powder), and strontium oxide (Alfa Aesar, 99.5%, metals basis) were added to glass batch in the percentages of 0.5, 1, 1.5, and 2.5 by decreasing SiO_2_ and CaO amount. The antibacterial glass compositions are given in Table 2.

After determining the antibacterial glass compositions, 12 glass batches of about 120 g were prepared. The batches were melted at 1450 °C for 3 h using a platinum crucible followed by annealing at 550 °C for 1 h. After annealing, the glass samples seen in Figure 1 were obtained.

### 2.2. Characterization of Antibacterial Glasses

Antibacterial properties of glasses were evaluated against a Gram-negative bacteria (*Escherichia coli* ATCC 25,922) according to the ISO 22196: 2011 standard method (Measurement of antibacterial activity on plastics and other nonporous surfaces). The control and test samples (50 mm × 50 mm) were initially cleaned and sterilized under UV irradiation for 2 h (each side for 1 h) and then inoculated with 100 μL of Mueller Hinton broth with 0.5% agar containing 2.5−10 × 10^5^ cells/mL (Merck KGaA, Darmstad, Germany). Then, test inoculum was covered with a piece of sterile film (40 mm × 40 mm). The film was used to cover and to spread inoculum. After incubation at 35 °C for 24 h, sterile film together with agar was removed and placed into the phosphate buffer saline (PBS) solution. Enumeration of survivals was performed by pour plate culture method on plate count agar after incubation at 35 °C for 48 h. The number of viable bacteria for test and control samples were calculated using following equation, where *N* is the number of viable bacteria recovered per cm^2^ per test specimen; *C* is the average plate count for duplicate plates; *D* is the dilution factor for the plates counted; *V* is the volume, in mL, of PBS added to the specimen; and *A* is the surface area, in mm^2^, of the cover film. Each analysis was performed in quadruplicate.
(1)$$N=\left(100\times C\times D\times D\times V\right)/A$$

Inactivation ratio (%) in bacterial count was calculated using following equation [31].
(2)$$Inactivation\;ratio\;\left(\%\right)=100\;\times \;\frac{N_{control}-N_{test}}{N_{control}}$$

The release of ions from glasses (ISO 6486) was measured as a function of immersion time in acetic acid and water with the aid of inductively coupled plasma-mass spectrometer (ICP-OES, Perkin Elmer Avio 200, Waltham, Massachusetts, USA). The antibacterial glasses were tested according to 84/500/EEC Directive and BS 6748. According to the 84/500/EEC Directive and BS 6748 standard, the release of lead (Pb) and cadmium (Cd) from the inner surface of the glassware intended to come into contact with foodstuffs using 4% (*v*/*v*) acetic acid at 22 ± 2 °C and during 24 ± 0.5 h was determined by ICP-OES.

The amorphous structure of the obtained glasses was identified using an X-ray diffraction analyzer (PANalytical Empyrean XRD, Malvern, UK, 45 mA, 40 kV, Scan range: 10°–60°, Step size: 0.013°).

The energy levels of silver oxide, which is effective in antibacterial properties of glass, were determined by XPS analysis. Thermo Scientific K-Alpha spectrometer (Thermo Fisher Scientific, Waltham, MA, USA) was used for this analysis. The spectrometer has an aluminum anode (Al Kα = 1468.3 eV) at an electron takeoff angle of 90° (between the sample surface and the axis of the analyzer lens). A flood gun was used to avoid charging. Accelerated Ar ions at 3000 eV was used for 30 s to clean the top surface from any organic impurities. The spectra were recorded using an Avantage 5.9 data system.

PE Lambda 900/950 UV-VIS-NIR Spectrophotometer (Perkin Elmer, Waltham, MA, USA) was used to determine the transmittance (% T), absorption (A) and reflection (% R) measurements of reference (nonantibacterial typical container glass) and antibacterial glasses. This device is a computer-controlled, double-beamed, dual-monochromator type spectrophotometer that is used to determine transmittance, absorption, and reflection measurement values in the ultraviolet–visible region–near infrared region of the spectrum, 185–3200 nm (nanometer) in the spectral range of 200–2500 nm when using the collector spheres. The UV WinLab software program, which is used to operate this device, allows the use of 4 different methods.

The densities of reference (nonantibacterial typical container glass) and antibacterial glasses were measured by Mettler Toledo Density Kit (Mettler Toledo, OH, USA) at room temperature using the principle of Archimedes and water as buoyancy liquid.

The thermal expansion coefficients of the reference and antibacterial glasses were detected using a dilatometer (NETZSCH DIL 402 PC, NETZSCH, Selb, Germany).

Hardness and reduced elastic modulus values were measured with nanoindenter (M1, NANOVEA, Irvine, CA, USA). The indentations were performed to maximum load of 300 mN at loading and unloading rates 600 mN/mN. Berkovich tip calibrated on fused silica was used for the indentation, and 10 indents were conducted on each sample.

ISO 695, “Glass Resistance to attack by a boiling aqueous solution of mixed alkali—method of test and classification” standard, was applied in order to determine and classify the alkali strength of the glass samples. This method involves determining and classifying the resistance of the glass samples interacting with the sodium carbonate and sodium hydroxide aqueous solution boiling at 102.5 ± 0.5 °C for 3 h according to the mass loss on the unit surface.

### 2.3. High Temperature Melting Observation

Melting and fining properties of glass composition (2% Ag_2_O) were also investigated using High Temperature Melting Observation System (HTMOS). As seen in Figure 2, the prepared batch was melted in a silica tube and video was recorded by the camera in the system from the beginning to the end of the melting process. Besides these transactions; aid of quantities of CO_2_ and SO_2_ gases, which are important for melting and fining, were measured using FTIR (Fourier Transform Infrared).

## 3. Results and Discussions

### 3.1. Antibacterial and Ion Release Tests of the Obtained Glass Samples

CuO- and SrO-doped glasses did not show any antibacterial activity according to the antibacterial test results. However, an antibacterial effect was observed for the Ag_2_O-doped glasses. Therefore, the studies continued with the Ag_2_O-doped glasses. When the antibacterial test results of Ag_2_O-doped glasses, given in Figure 3, were examined, it was found that the glass sample doped with 2.5% Ag_2_O has the best antibacterial effect.

No bacterial colony was observed in the test of 2.5% Ag_2_O against *Escherichia coli* in all replicates. According to the results, inactivation rate for *E. coli* of 2% and 2.5% Ag_2_O-containing glasses was higher than 99.99% (>5 log). Esteban-Tejeda et al. [32,33] reported that a logarithm reduction higher than 3 means safe disinfection and high antimicrobial activity. Moreover, International Microbiological Criteria for Dairy Products suggested that inactivation rate for *E. coli* must be 99.9% in order to be account as antibacterial glassware [34]. As seen in Figure 3, inactivation rate for *E. coli* of 2% and 2.5% Ag_2_O-containing glasses is 99.99%, which is the limit criteria for *E. coli* according to International Microbiological Criteria for Dairy Products [34].

Guldiren and Aydin provided antibacterial properties to their soda lime glass by adding silver and copper separately and together. Unlike the conventional melting method, the ion exchange method was used in their study. It means that the conventional soda lime glass production needs extra process. According the antimicrobial test results, *E. coli* bacteria decreased to 99.882% as the maximum. It can be seen that antibacterial soda lime glass obtained in our study by conventional melting method has better antibacterial activity against to *E. coli* than antibacterial soda lime glass obtained by ion exchange. It is also important to note that soda lime antibacterial glass obtained by conventional melting method has more advantageous in terms of time and cost compared with those obtained by ion exchange method because it does not need an extra process [31,35].

Esteban-Tejeda et al. used silver and copper nanoparticles in antibacterial soda lime glass studies. In their study, they mixed the soda lime glass with silver and copper nanoparticles and then sintered the obtained powders. In the results obtained in these studies, the reduction rate of bacteria is lower than this study [32,33].

Apart from these studies, sol–gel technique was used to give antibacterial feature to the soda lime glasses. Lee et al. showed that *E. coli* bacteria decreased to 99.99995% in the soda lime glass coated with silver-doped antibacterial film [36]. Since sol–gel method has disadvantages like long processing time, high cost of the process, and stripping of coating from the glass surface, producing antibacterial soda lime glass using melting method is much more advantageous.

After the results of the antibacterial test, toxic results of glass samples doped with 2.0% Ag_2_O and 2.5% Ag_2_O were evaluated using the values obtained from the ICP-OES test results. The test was performed in two replicates. One of the samples was kept in 4% (*v*/*v*) acetic acid for 1 day and the other sample was kept in 4% (*v*/*v*) acetic acid for a week at 22 °C (±2). As can be seen from Table 3, the released amounts of the metal ions from the obtained glasses were below toxic values for both acute toxicity and chronic effects of silver exposure. [37,38] The amount of released silver ions is so high with respect to the release in a week because mass transfer at the first moment occurs rapidly due to the concentration difference with the dissolution in the glass [39]. According to the antibacterial and ion release test results, glass sample doped with 2% Ag_2_O was selected for further studies in order to reduce the cost of the raw materials for the glass production.

### 3.2. Material Characterization

First of all, XRD analysis was performed on the obtained glass sample to detect the phase structure. It was seen from Figure 4 that the glass doped with 2% silver oxide was fully amorphous.

In addition to XRD analysis, XPS analysis was performed to find the energy levels of silver oxide. As seen from Figure 5a, the most prominent peaks belong to oxygen and silica. The peaks in the range of 360–380 eV belong to the silver (I) oxide, and the expanded graph of this region can be seen from Figure 5b. XPS analysis results indicated that silver (I) oxide was successfully incorporated into the glass doped with 2% Ag_2_O. This result also confirms the antibacterial property of the glass sample since silver oxide was detected on the surface of the glass sample.

Alkali strength test results showed that the weight loss of the glass sample containing 2% silver oxide was 68.37 mg/dm^2^. It was detected that alkali resistance class of glass is A1 according to Alkali Resistance Test Limits, which showed that alkali resistance characteristic is in low level degradation.

Furthermore, some tests were performed on the antibacterial glass sample to see physical and optical properties of the reference (nonantibacterial typical container glass) and antibacterial (2% Ag_2_O) glasses. As can be seen from Table 4, the physical properties of antibacterial glass and reference glass (nonantibacterial typical container glass) are close to each other, but optical properties of two glasses are different from each other. The color of the silver-doped glass is amber, which is totally different from the soda lime glass.

Viscosity measurements of the glasses were performed to see preliminary melting behavior. According to the results given in Table 5, viscosity values of the reference (nonantibacterial typical container glass) and the antibacterial glasses are slightly different, but when the softening temperature is reached, this difference is seen to decrease.

Lastly, hardness and elastic modulus of the glasses were measured to see mechanical behavior. The results showed that the hardness of antibacterial glass was 5.5 ± 0.1 GPa and that this value was the same as hardness of reference (nonantibacterial typical container glass) glass. Elastic modulus of antibacterial glass was measured as 73.2 ± 0.9 GPa, while the elastic modulus of reference glass was 73.3 GPa. It means that elastic modulus of two glasses is nearly same. Overall results indicated that the glass sample doped with 2% Ag_2_O has viscosity and optical and mechanical properties similar to those of commercial soda lime glass, which are glass types consisting of silicon dioxide, sodium oxide, calcium oxide, and magnesium oxide (MgO), and is generally used for the production of flat glass, container glass, and glassware [40,41].

### 3.3. High Temperature Melting Observation System

One of the originalities of this study is performing the High Temperature Melting Observation System (HTMOS) on the Ag_2_O-doped antibacterial soda lime glass obtained by conventional melting method for the first time. Figure 6a shows CO_2_ release during HTMOS measurements. As can be seen from Figure 6b that the melting reactions of antibacterial composition doped with 2% Ag_2_O started later than that of reference composition. In addition, according to Video 1 and Video 2 (See Appendix A), the fining of the antibacterial composition doped with 2% Ag_2_O began slower than that of the reference glass (nonantibacterial typical container glass but also appeared to be better than the reference glass’s fining (nonantibacterial typical container glass), which can be seen from the surface of both glasses at the end of the experiment. The reason of that is the gases can reach the glass surface more easily and can be thrown out of the glass when viscosity of the glass is low.

Arslan et al. and Gulin et al. made detailed melting behavior experiments of different soda lime glass compositions using High Temperature Melting Observation System. According to the results, it can be seen that antibacterial soda lime glass obtained by conventional melting method shows a similar melting behavior to these compositions [42,43]. This result indicated that the antibacterial glass composition can be produced in soda lime glass furnaces without changing any furnace design and production parameters.

## 4. Conclusions

In this study, soda lime glass with antibacterial properties was produced with the addition of silver oxide to the batch. According to ICP-OES results, migration values were below toxic values. The antibacterial property of the glass produced with 2% silver oxide was found to be quite good, even with spores on glass with 2.5% silver oxide added glass. In addition, the degradation rate of antibacterial glass (2% Ag_2_O) was at a low rate against the alkali solution and identified as class A1. The physical properties of antibacterial glass and reference glass (nonantibacterial typical container glass) were almost the same, but their optical properties were different. Viscosity measurements indicated that the antibacterial glass was more viscous than that of the reference glass (nonantibacterial typical container glass). Overall results showed that the antibacterial composition can be adapted to glass industry without any need of change in the terms of furnace design and production parameters.

## Figures and Tables

**Figure 1 materials-13-04827-f001:**
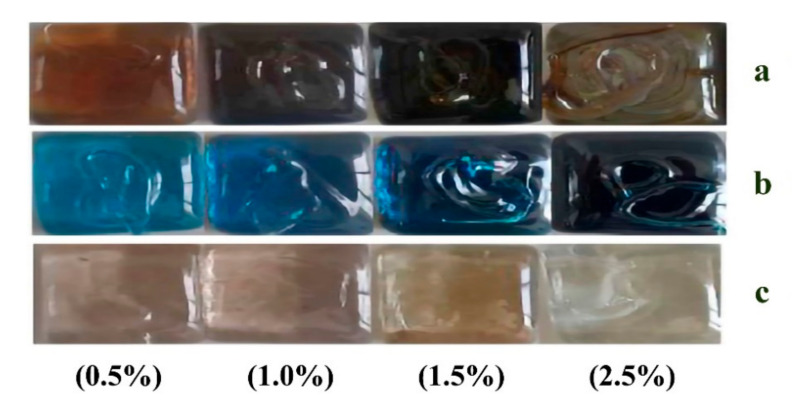
Glass samples containing (**a**) silver (I) oxide, (**b**) copper (II) oxide, and (**c**) strontium oxide.

**Figure 2 materials-13-04827-f002:**
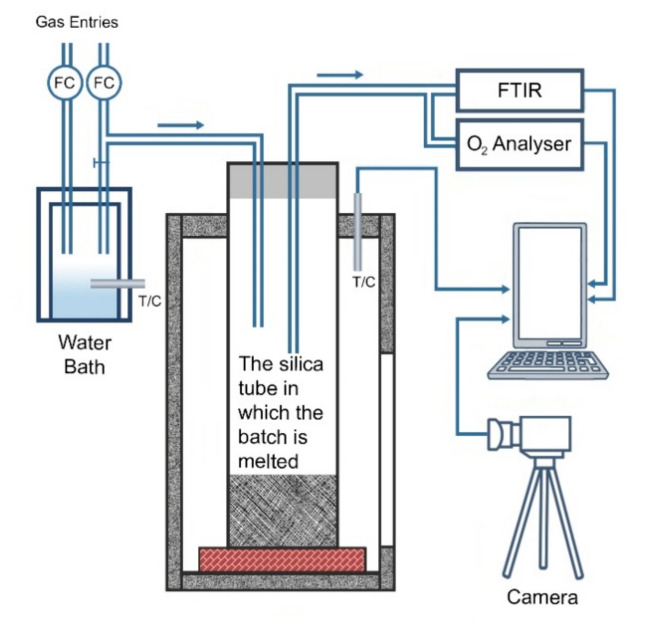
High Temperature Melting Observation System.

**Figure 3 materials-13-04827-f003:**
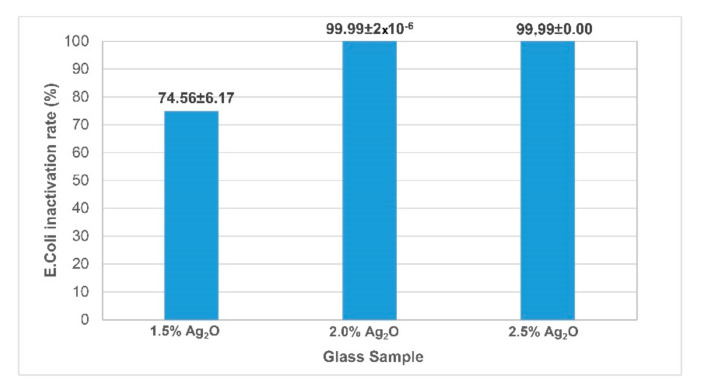
Inactivation rate of *Escherichia coli*.

**Figure 4 materials-13-04827-f004:**
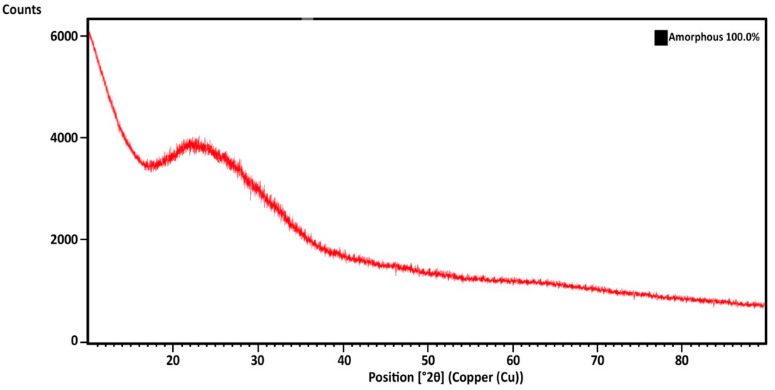
XRD results.

**Figure 5 materials-13-04827-f005:**
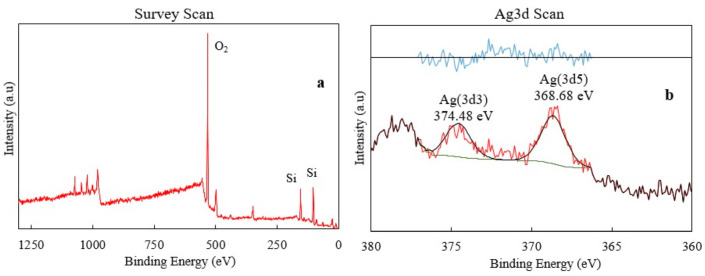
(**a**) XPS analysis and (**b**) silver energy range graph of glass sample doped with 2% Ag_2_O.

**Figure 6 materials-13-04827-f006:**
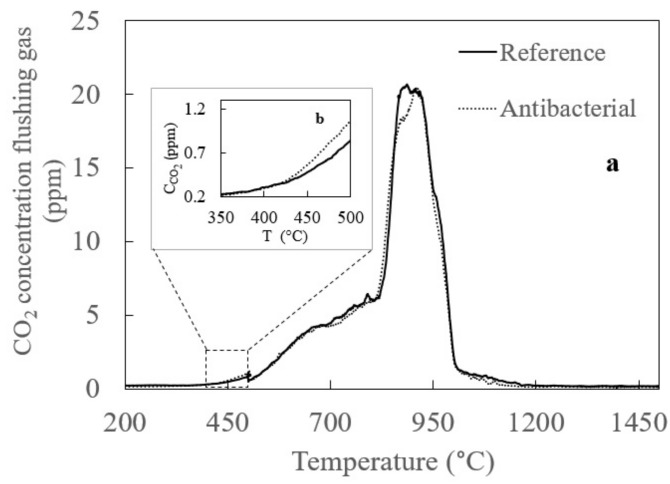
(**a**) CO_2_ release during HTMOS measurements and (**b**) CO_2_ release at the start of melting reaction.

**Table 1 materials-13-04827-t001:** Composition of typical glass container (nonantibacterial) glass composition.

SiO_2_ (%)	Al_2_O_3_ (%)	Fe_2_O_3_ (%)	TiO_2_ (%)	CaO (%)	MgO (%)	Na_2_O (%)	K_2_O (%)	SO_3_ (%)
71.40	1.71	0.06	0.06	9.79	3.28	13.17	0.31	0.24

**Table 2 materials-13-04827-t002:** Compositions of antibacterial glasses.

SiO_2_ (wt.%)	Al_2_O_3_ (wt.%)	Fe_2_O_3_ (wt.%)	TiO_2_ (wt.%)	CaO (wt.%)	MgO (wt.%)	Na_2_O (wt.%)	K_2_O (wt.%)	SO_3_ (wt.%)	Ion (wt.%)
71.40	1.71	0.06	0.06	9.29	3.28	13.17	0.31	0.24	0.50
70.90	1.71	0.06	0.06	9.29	3.28	13.17	0.31	0.24	1.00
70.40	1.71	0.06	0.06	9.29	3.28	13.17	0.31	0.24	1.50
69.40	1.71	0.06	0.06	9.29	3.28	13.17	0.31	0.24	2.50

**Table 3 materials-13-04827-t003:** Inductively coupled plasma-mass spectrometer (ICP-OES) test results.

	The Released Amount of Silver Ion (Glass Doped with 2% Ag_2_O)	The Released Amount of Silver Ion (Glass Doped with 2.5% Ag_2_O)
A day in acetic acid	0.28	0.59
A day in water	0.16	0.22
A week in acetic acid	0.72	0.89
A week in water	0.33	0.43

**Table 4 materials-13-04827-t004:** Physical and optical test results.

	Reference	Antibacterial
**Thermal expansion coefficient (10^−7^/°C)**	86.5	85.5
**Density (g/cm^3^)**	2.493	2.512
**Refractive index**	1.5200	1.5205
**Color parameters (standard 3 mm)**		
**Brightness (%)**	72.0	19.7
**Dominant wavelength (nm)**	556.5	588.7

**Table 5 materials-13-04827-t005:** Viscosity measurement results.

Viscosity	Temperature (°C)		
	Reference	Antibacterial (First Measurement)	Antibacterial (Second Measurement)
**log η = 2.25 (±0.018) (melting temperature)**	1372	1388	1388
**log η = 2.50 (±0.014)**	1309	1323	1323
**log η = 2.75 (±0.009)**	1252	1265	1264
**log η = 3.00 (±0.011) (Gob temperature)**	1200	1212	1211
**log η = 3.25 (±0.012)**	1154	1162	1162
**log η = 3.50 (±0.015)**	1111	1119	1118
**log η = 4.00 (±0.016)**	1036	1042	1041
**log η = 7.65 (softening temperature)**	734 (±2.3)	739 (±2.3)	–
**Working range (WR) T_logη=3_ − T_logη=7.65_**	466	473	–

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
