# Peer review of "Production of Soda Lime Glass Having Antibacterial Property for Industrial Applications"

_materials, 2020, doi:10.3390/ma13214827_

Round 1

Reviewer 1 Report

The topic of the paper "Production of soda-lime glass having antibacterial property..." is of interest for industrial application in food industry and others.

However some details should be improved before the paper is acceptable for publication.

In details:

English language must be improved, above all as long as sentences structure is concerned. The authors seem not to know how to use coma, and more in general punctuation, in sentences.

Line 73 - The authors should explain the meaning of this statement.

Line 163, Table 3 - The authors should deepen their discussion about the amount of released silver ions, explaining why the initial release (after one day soaking) is so high with respect to the release in a week. 

Author Response

Response to Reviewer 1 Comments

We sincerely appreciate all valuable comments and suggestions, which helped us to improve the quality of the article.

Point 1:

English language must be improved, above all as long as sentences structure is concerned. The authors seem not to know how to use coma, and more in general punctuation, in sentences.

Response 1:

The text was revised throughout the manuscript by a native speaker.

Point 2:

The authors should explain the meaning of this statement (Line 73).

Response 2:

Thanks for the notice about the statement. It was explained in detail in the revised manuscript.

Point 3:

The authors should deepen their discussion about the amount of released silver ions, explaining why the initial release (after one day soaking) is so high with respect to the release in a week (Line 163, Table 3).

Response 3:

The reason of high initial release (after one day soaking) was explaind in the revised manuscript and a reference was added about this explanation.

Reviewer 2 Report

The manuscript provide original study of antibacterial glasses. Nevertheless, some of the paragraphs of the article must be, in my opinion, be improved before publication:

1) Fig. 1: The sample with 2.5% of Ag2O seems to be heterogenous (pale and dark lines are visible). Did it not have an impact on glass properties? 

2) Please provide all microorganisms names in italic. 

3) I see some inconsequence in the results presentation. The authors mentioned samples with 0.5%, 1.0%, 1.5%, 2.0% and 2.5% of Ag2O. However, the Fig. 1 showed only the samples 0.5%, 1.0%, 1.5% and 2.5%, in turn Fig. 3. showed the results obtained for 1.5%, 2.0% and 2.5%. Similarly Fig. 4 and Table 3 did not show results for all Ag2O concetrations. What is more important, the results for control sample are not shown in any case.

4) The antibacterial properties test is described unsufficiently, among others, the origin of the strain and its exact name (i.e. database number of given strain) should be provided. Moreover, the incubation condition, sterilization procedure of glasses before experiments, the liquid medium composition, cells' counting method must be described. 

5) Fig. 3: the error bars are lacking. Without them all discussion and conlusions about the antibacterial properties is unfortunately unjustified. Moreover, if the used methods provide the accuracy llowing to describe results with two or three decimal places?

6) The quality of Fig. 5 is insufficient. Moreover, its desctiption and discussion is rather laconic.

7) Table 5. Could the Authors explain the necessity of two measurements for antibacterial samples?

8) The quality of Fig. 6 is also insufficient.

Author Response

Response to Reviewer 2 Comments

We sincerely appreciate all valuable comments and suggestions, which helped us to improve the quality of the article.

Point 1:

Fig. 1: The sample with 2.5% of Ag2O seems to be heterogenous (pale and dark lines are visible). Did it not have an impact on glass properties?

Response 1:

The sample with 2.0% of Ag2O seems to be heterogenous like 2.5% of Ag2O. Properties of glass sample with 2.0% of Ag2O were measured and it was found that there was no significant effect of its heterogenous appearance on the properties of the glass sample as it can be seen from Tables 4 and 5.

Point 2:

Please provide all microorganisms names in italic.

Response 2:

Thanks for the notice about microorganisms names. All names were written in italic in the revised manuscript.

Point 3:

I see some inconsequence in the results presentation. The authors mentioned samples with 0.5%, 1.0%, 1.5%, 2.0% and 2.5% of Ag2O. However, the Fig. 1 showed only the samples 0.5%, 1.0%, 1.5% and 2.5%, in turn Fig. 3. showed the results obtained for 1.5%, 2.0% and 2.5%. Similarly Fig. 4 and Table 3 did not show results for all Ag2O concentrations. What is more important, the results for control sample are not shown in any case.

Response 3:

Antibacterial tests were not applied glass samples doped with 0.5% and 1.0% Ag2O since antibacterial activity of glass with 1.5% Ag2O was below the limit criteria for E.Coli according to International Microbiological Criteria for Dairy Products. Therefore, ion release studies conducted only glass samples doped with 2 % Ag2O and 2.5 % Ag2O as can be seen from Table 3.  According to the antibacterial and ion release test results glass sample doped with 2 % Ag2O was selected for further studies in order to reduce the cost of the raw materials for the glass production as it was stated in the manuscript. Therefore, material characterization studies continued with only the glass sample doped with 2 % Ag2O.

Point 4:

The antibacterial properties test is described unsufficiently, among others, the origin of the strain and its exact name (i.e. database number of given strain) should be provided. Moreover, the incubation condition, sterilization procedure of glasses before experiments, the liquid medium composition, cells' counting method must be described.

Response 4:

The antibacterial properties test was described in detail in the revised manuscript.

Point 5:

Fig. 3: the error bars are lacking. Without them all discussion and conlusions about the antibacterial properties is unfortunately unjustified. Moreover, if the used methods provide the accuracy llowing to describe results with two or three decimal places?

Response 5:

Thanks for the notice about error bars. The error bars were added into Figure 3 in the revised manuscript.

Point 6:

The quality of Fig. 5 is insufficient. Moreover, its desctiption and discussion is rather laconic.

Response 6:

Figure 5 was revised according to the publication standard.

Point 7:

Table 5. Could the Authors explain the necessity of two measurements for antibacterial samples?

Response 7:

Because of the deviation in viscosity measurements two measurements were made on antibacterial glass sample.

Point 8:

The quality of Fig. 6 is also insufficient.

Response 8:

Figure 6 was revised according to the publication standard.

Round 2

Reviewer 2 Report

Thank the Authors for their kind response. I have only one remark - the Figures 1, 2 and 3 are of quite poor quality (e.g. the pixels re easily visible). I suggest to improve them.

Author Response

Response to Reviewer 2 Comments

We sincerely appreciate valuable suggestion, which helped us to improve the quality of the article.

Point 1:

I have only one remark - the Figures 1, 2 and 3 are of quite poor quality (e.g. the pixels re easily visible). I suggest to improve them.

Response 1:

Figure 1, 2 and 3 were revised according to the publication standard.
